# The impact of visually simulated self-motion on predicting object motion

**Björn Jörges** [ID]*, **Laurence R. Harris** [ID]

Center for Vision Research, York University, Toronto, Ontario, Canada

* bjoern_joerges@hotmail.de

This is a Registered Report and may have an associated publication; please check the article page on the journal site for any related articles.

**Data Availability Statement:** https://github.com/b-jorges/Predicting-while-Moving/tree/main/Data.

## Abstract

To interact successfully with moving objects in our environment we need to be able to predict their behavior. Predicting the position of a moving object requires an estimate of its velocity. When flow parsing during self-motion is incomplete–that is, when some of the retinal motion created by self-motion is incorrectly attributed to object motion–object velocity estimates become biased. Further, the process of flow parsing should add noise and lead to object velocity judgements being more variable during self-motion. Biases and lowered precision in velocity estimation should then translate to biases and lowered precision in motion extrapolation. We investigated this relationship between self-motion, velocity estimation and motion extrapolation with two tasks performed in a realistic virtual reality (VR) environment: first, participants were shown a ball moving laterally which disappeared after a certain time. They then indicated by button press when they thought the ball would have hit a target rectangle positioned in the environment. While the ball was visible, participants sometimes experienced simultaneous visual lateral self-motion in either the same or in the opposite direction of the ball. The second task was a two-interval forced choice task in which participants judged which of two motions was faster: in one interval they saw the same ball they observed in the first task while in the other they saw a ball cloud whose speed was controlled by a PEST staircase. While observing the single ball, they were again moved visually either in the same or opposite direction as the ball or they remained static. We found the expected biases in estimated time-to-contact, while for the speed estimation task, this was only the case when the ball and observer were moving in opposite directions. Our hypotheses regarding precision were largely unsupported by the data. Overall, we draw several conclusions from this experiment: first, incomplete flow parsing can affect motion prediction. Further, it suggests that time-to-contact estimation and speed judgements are determined by partially different mechanisms. Finally, and perhaps most strikingly, there appear to be certain compensatory mechanisms at play that allow for much higher-than-expected precision when observers are experiencing self-motion–even when self-motion is simulated only visually.

## Introduction

We are constantly immersed in complex, dynamic environments that require us to interact with moving objects, for example when passing, setting, or hitting a volleyball, or when

**Funding:** The author(s) received no specific funding for this work.

**Competing interests:** The authors have declared that no competing interests exist.

deciding whether we can make a safe left turn before another car reaches the intersection. In such situations, it can often be important to predict how objects in our environment will behave over the next few moments. Predictions allow us, for example, to time our actions accurately despite neural delays [1–3] in perceiving moving objects and issuing and executing motor commands [4,5]. Delays of between 100ms and 400ms between visual stimulation and motor response are generally assumed [6]. Without predicting or anticipating motion, we would thus always be acting on outdated positional information and be doomed to fail when attempting to intercept moving objects, particularly when they are relatively fast. Further, and perhaps more obviously, predictions are also important when moving objects are occluded during parts of their trajectory [7–9], or when the observer has to avert their eyes or turn away from the target.

When an observer is moving while attempting to interact with moving objects in their environment, further difficulties arise. Even in the simplest case, when the observer has a good view of the target and predictions are only necessary to compensate for neural delays, the visual system needs to separate retinal motion created by observer motion from retinal motion due to object motion in order to judge an object's trajectory accurately. A prominent hypothesis on how this is achieved is the Flow Parsing hypothesis [10–16]. This hypothesis states that to solve this problem humans parse optic flow information and subtract this visual stimulation attributed to self-motion from global retinal motion. The remaining retinal motion can then be scaled with an estimate of the distance between the observer and the moving object [17] to obtain the world-relative velocity of the object. While this process was originally posited as a purely visual phenomenon [18,19], more recent studies have shed some light on the multisensory nature of this process: Dokka and her colleagues [20] found, for example, that compensation for self-motion in speed judgements was more complete when both visual and vestibular information indicated self-motion than when either visual or vestibular cues indicated that the observer was at rest. They further found precision to be lower in the presence of self-motion, which they attributed to self-motion information being noisier than optic flow information. Subtracting noisy self-motion information [21] from less noisy optic flow [20] would then make the estimate of object motion noisier in a moving observer than in a static observer. It is important to note that such a flow parsing mechanism should be active even while the participant at rest. The assumption here is that the noise added during flow parsing is proportional to the self-motion speed (in a Weber's Law-like fashion), that is, when the observer is not moving at all the added noise is minimal, whereas higher self-motion speeds lead to more noise.

Several studies have shown that flow parsing is often incomplete [9,22–27]. A candidate explanation for this incompleteness is an inaccurate estimation of self-motion: typically, studies have used stimuli where only some cues indicated that the observer was moving (usually visual and/or vestibular cues), while efferent copy cues indicated that their bodies were at rest. If all self-motion cues are integrated after weighting them (e.g., according to their relative reliabilities [21,28,29]), this would then lead to an underestimation of self-motion, which is consistent with the biases found in the studies cited above. Further, while we did not find evidence for an effect of self-motion on precision in a recent study [30], we believe it is likely due to the fact that the effect was noisier than anticipated, resulting in a lack of statistical power.

Fig 1 shows a simple schematic of the processes we assume to be at play when predicting object motion during self-motion: the organism first estimates its own motion in the environment from the various sources of information available to it. Based on this self-motion estimate, the organism then makes a prediction about the retinal motion that this self-motion would be expected to create. The predicted retinal motion is then subtracted from the total observed retinal motion and any remaining motion is attributed to the object, a process we call "multisensory flow parsing" to distinguish it from the purely visual conceptualization of flow

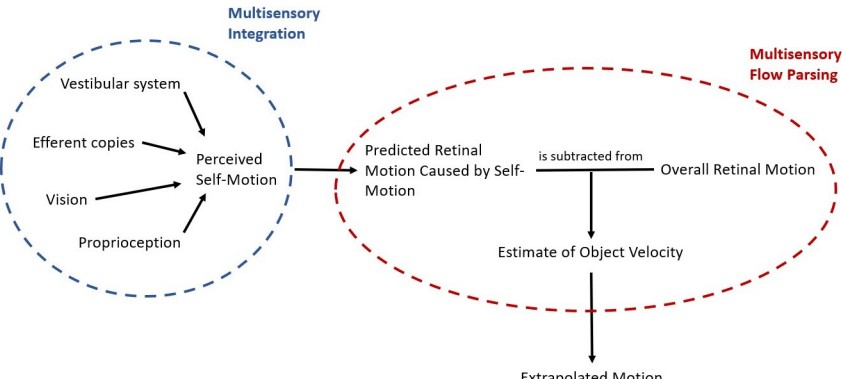

**Fig 1. Schematic of the processes at play when predicting motion during self-motion.** First, an estimate of the self-motion velocity is obtained by integrating the available cues from different sensory sources. This velocity estimate is used to predict the retinal motion that would be caused by self-motion. Finally, an estimate of the physical object velocity is obtained by subtracting the predicted retinal motion due to self-motion from the global retinal motion and the further trajectory of the object is extrapolated based on this estimate.

parsing brought forward by authors like Wertheim [15,16] or, more recently, Dupin and Wexler [10]. This step adds noise to the velocity estimate because the self-motion estimate is noisier than the retinal input [20]. The further trajectory of the object would then be extrapolated based on this estimate of its velocity.

This schematic provides an overview over the general mechanism that might be at play when predicting future motion of an object observed during self-motion. Depending on the motion profiles of observer and object further complications may arise. For example, retinal speeds depend not only on the physical speeds and directions of the observer and object but also on the distance between them and can therefore change systematically even when observer and object move at constant physical speed without changing direction. To obtain a veridical representation of the physical velocity of the object, the observer thus has to perform additional computations, including estimating their distance to the object, the direction of the object in relation to their own direction of motion, and the necessary transformations to obtain the physical object speed from these values [17].

Some studies suggest that biases incurred while estimating motion, e.g., due to the Aubert-Fleischl effect which lowers the perceived speed of a target when an observer tracks it with their gaze [31], or due to low contrast in the stimulus [32], might transfer to biases in motion extrapolation based on these speed estimates. It seems straight-forward that any biases and precision differences observed in the perception of speed would correlate perfectly with errors and precision differences in time-to-contact judgements. However, there are two complications: first, participants might integrate biased and less precise speed information obtained during self-motion with prior information they have formed in response to previous exposure to the stimulus. They might thus extrapolate motion biased on a combination of prior information and (biased and more variable) online information. Further, it has been reported that under certain circumstances perceptual judgements and action-related tasks can be based on separate cues (see, e.g., [33–35]). While it remains an appealing hypothesis, it should thus not be assumed that biases and variability differences in time-to-contact judgements reflect only biases and variability differences in speed estimation. Studying to what extent biases and variability differences in online information acquired while viewing a target influence the way we extrapolate its further motion will help us better understand the predictive mechanisms at play

not only when the target is occluded or the observer averts their gaze from the target but also when timing interceptive actions accurately despite unavoidable neural delays [36]. In the present study, we therefore investigate how biases in speed estimation elicited by visual self-motion impact the prediction of object motion.

More specifically, we test three interconnected hypotheses:

- Predictions of where a moving object will be in the future will be biased (Hypothesis 1a) and more variable (Hypothesis 1b) in the presence of visually simulated self-motion.

- Object speed judgements will be biased (Hypothesis 2a, which constitutes a replication of an earlier result of Jörges & Harris, 2021) and more variable (Hypothesis 2b, a more highly powered follow-up to a hypothesis for which Jörges & Harris, 2021 did not find significant support) in the presence of visually simulated self-motion.

- The effect of visually simulated self-motion on motion extrapolation can be predicted from its effect on speed estimation, both in terms of bias (Hypothesis 3a) and its variability (Hypothesis 3b).

## Methods

### Apparatus

We programmed all stimuli in Unity 2020.3.30f1. Given the on-going COVID-19 pandemic, some participants who were owners of head-mounted VR devices (HMDs) were tested in the safety of their own homes. Our experiment was compatible with all major modern HMDs. To minimize variability introduced by the use of different HMDs, each program we used to present stimuli limited both the horizontal and the vertical field of view to 80˚. We didn't expect there to be relevant differences in frame rate, as Unity caps the frame rate at 60 Hz. Our programs were, further, much less demanding in processing power than any VR application remote participants were likely to run; that is, frame rate dropping below 60 Hz should occur almost never. Participants tested at our facilities at York University were tested with a VIVE Pro Eye.

### Participants and recruitment

We recruited three participants (n = 3) with HMDs in their possession online for them to perform the experiment in their homes. Recruitment occurred through social media (such as Twitter, Facebook, and Reddit). For the remaining n = 37 participants, testing occurred at York University, following all applicable guidelines for safe face-to-face testing. Remote participants received a monetary compensation of 45 CAD for participation in the experiment; participants recruited through the York University participant pools received course credit instead of a monetary compensation. All participants were screened for stereo-blindness with a custom Unity program (downloadable on GitHub: https://github.com/b-jorges/Stereotest) in which participants had to distinguish the relative depth of two objects that were matched in retinal size. Participants were only included if they answer correctly on 16 out of 20 trials. The simulated disparity was 200 arcsec. Participants who repeated failed the VR-based stereo-test but had no history of issues with their stereovision were assessed using an analogue stereo fly test, for which all participants achieved or exceeded a stereoacuity of 100 arcseconds as measured per the circles test. While this allowed only for a coarse assessment of the participants' stereovision, our experiment was not critically dependent on a high stereoacuity. We tested 20 men and 20 women (see Power Analysis) and no non-binary participants or participants of

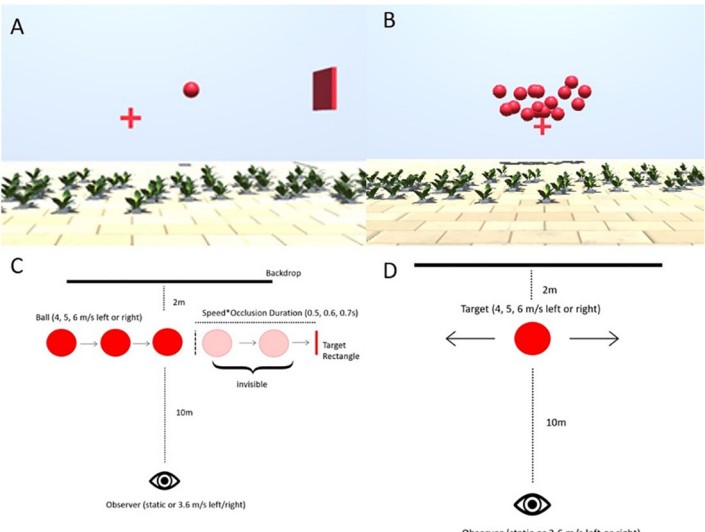

**Fig 2.** **A**. Screenshot from the prediction task while the ball was visible. **B**. Screenshot from the speed estimation task while the ball cloud was presented. **C**. Schematic of the prediction task. **D**. Schematic of the speed estimation task. E. An illustration of the time course of target speed (red line, constant at 4, 5 or 6 m/s across the 0.5s trajectory), the self-motion speed (grey, ramping up over the first 50 ms and ramping down over the last 50 ms) and the relative speed between target and observer (orange). The shaded areas represent the ramp-on and ramp-off of the self-motion speed.

other genders signed up. The experiment was approved by the York University Human Participants Review Committee and was conducted in accordance with the Declaration of Helsinki. Data collection began on June 1st, 2022, and was completed on September 18th, 2023. All participants gave their written informed consent.

## Stimulus

Each participant performed two main tasks in an immersive VR environment: a prediction task and a speed estimation task. Every participant completed both tasks, and we counterbalanced the order in which they completed them such that 20 participants (10 women and 10 men) started with the prediction task, and 20 participants (10 women and 10 men) started with the speed estimation task. All programs we used to present the stimuli are available on Open Science Foundation (https://osf.io/gakp5/) and the Unity projects can be downloaded on Open Science Foundation (https://osf.io/6mz4w/).

For both tasks, we displayed a circle in the middle of the observer's field of view that moved with their head rotation in front of the Earth-stationary simulated world. Participants were instructed to keep this circle surrounding the fixation cross. When the center of the circle was within 2.5˚ (vertically and horizontally) of the fixation cross, it disappeared to indicate to the participant that their head was positioned correctly. We further recorded head position whenever the ball or the ball cloud (see below) were visible. Since recording head position on each frame might slow down the program on older systems, we opted to record the mean rotation (horizontally and vertically) over bins of five frames.

**Prediction.** In the prediction task (see Fig 2A and see also this video on OSF (https://osf.io/rkg23/), we first showed participants a ball of 0.4 m diameter moving laterally 8m in front of them at one out of three speeds (4, 5, 6 m/s). We used this range of speeds in our previous study [37], while the size of the ball was diminished slightly in comparison to this study (see description of the speed estimation task for the rationale). The ball could travel to the left or to

the right. It appeared to the left of the observer when it traveled to the right, and on the right of the observer when it traveled to the left such that the visible part of the trajectory was centered in front of the observer. At the same time, a target rectangle was presented on the side towards which the ball was travelling. The ball disappeared after 0.5s and participants pressed the space bar on their keyboard in the moment they thought the ball hit the target. In order to curtail biases in speed perception due to the Aubert-Fleischl phenomenon [38], participants were asked to keep their eyes on a fixation cross that was presented straight ahead of them and slightly below the stimulus (see Fig 2A) and moved with the observer when they experienced visual self-motion. The target was presented at a distance that depended on the speed of the ball and the occlusion duration, which could be 0.5 s, 0.6 s, or 0.7 s. Speeds and occlusion durations were chosen such that, when the participant kept their gaze on the fixation cross, the whole trajectory (including the invisible part) unfolded within a field of view of 60˚, which was well within the effective field of view of any modern HMD. The distance between the point where the ball disappeared (the "point of disappearance"; see Fig 2C) and the target rectangle was given by the following equation:

$$Distance = Duration_{Occlusion} * Speed_{Ball} \tag{1}$$

While the target was visible, participants experienced lateral visual self-motion either in the same direction as the ball or in the opposite direction as the ball, or they remained stationary. The self-motion speed ramped up in a Gaussian fashion over the first 50 ms until it reached 4 m/s, then remained constant for 400 ms, and finally ramped down again over the last 50 ms before the ball became invisible. Overall, the observer moved 1.8 m over 500 ms (see Fig 2E for an illustration of the speeds at play in the different motion profiles). Please note that the different motion profiles elicited very different retinal speeds: observer motion in the opposite direction of the ball elicited higher retinal speeds overall than for a static observer or for observer motion in the same direction as the ball. Table 1 displays the mean absolute retinal speeds across the trajectory for all conditions. While our previous results [37] suggested that the role of the retinal speeds for the overall precision in motion estimation is subordinate to other sources of variability, retinal speeds are highly correlated with the expected effect of our self-motion manipulation on variability.

Participants completed a brief training of 18 trials before starting the main experiment (see this video on OSF: https://osf.io/4js5w/). The ball traveled at one of three speeds (2.5, 3.5, 4.5 m/s), going either left or right, and three occlusion durations (0.45, 0.55, 0.65 s). In the

**Table 1. Mean absolute retinal speeds across the visible part of the trajectory for each combination of ball speed and observer motion profile.** The script in which we derive these values can be found on GitHub (https://github.com/b-jorges/Predicting-while-Moving/blob/main/Geometry%20Prediction.R).

|  | Ball Speed | | |
|---|---|---|---|
|  | **4 m/s** | **5 m/s** | **6 m/s** |
| Observer Static | 22.2˚/s | 28.4˚/s | 34.0˚/s |
| Same Direction | 0.4˚/s | 5.9˚/s | 11.6˚/s |
| Opposite Directions | 44.4˚/s | 50.4˚/s | 55.8˚/s |

We further added a range of occlusion durations (0.1s, 0.2s, 0.3s, 0.4s, 0.8s, 0.9s, 1s) while the observer was static to get an estimate of how variability changed in response to different occlusion durations. Overall, participants completed 225 trials (3 ball speeds * 3 self-motion profiles * 3 occlusion durations * 5 repetitions + 3 ball speeds * 7 occlusion durations * 5 repetitions), which took around 10 minutes.

training, the ball reappeared upon pressing the spacebar in the position it would have been in at that moment. No visual self-motion was simulated in the training. This allowed participants to estimate their error (spatially) and helped them familiarize themselves with the task and the environment.

**Speed estimation.** In the speed estimation task (video on OSF: https://osf.io/xqkgy/), participants were presented with two motion intervals and had to judge which of them was faster. In one interval, they viewed a ball travelling to the left or to the right. As for the prediction task, this ball could have one of three speeds (4, 5, 6 m/s), and the participant could also experience visual self-motion in the same direction or in the opposite direction or remain static (see Fig 2D). Except for the self-motion intervals, the scene was static just like in the prediction experiment, and object motion occurred relative to the scene. The second motion consisted of a ball cloud of 2.5 m width and 1 m height at the same distance to the observer (see Fig 2B). Each ball in this cloud had the same diameter as the main target (0.4m) and balls were generated at one side of the cloud and then moved to the other side where they disappear. In our previous study [37], we used smaller balls for the ball cloud, which may have been a factor their judgements of speed being consistently overestimated relative to the single ball. We therefore decided to use the same ball size for both the single ball and the elements of the ball cloud. At any given moment, between 8 and 12 balls were visible. All the balls in the ball cloud moved at the same speed and were visible either until they reached the opposite side of the cloud area or until the motion interval ended after 0.5s observation time. The speed of these balls was constant throughout each trial and was governed by PEST staircase rules [39]. For each condition, we employed two pests: one started 30% above the speed of the single ball from the other motion interval, while the other started 30% below. The initial step size was 0.6 m/s and each pest terminated either after 37 trials, or when the participant had completed at least 30 trials and the step size dropped below 0.03 m/s. We modified the original PEST rules such that the step size was always twice the initial step size (that is, 1.2 m/s) for the first ten trials in order to spread out the values presented to the observer and allow for more robust JND estimates. We limited the range of speeds the ball cloud could take to between one third of the speed of the single ball and three times the speed of the single ball. Participants were asked to maintain fixation on the fixation cross that had the same characteristics as in the prediction task, that is, it was always presented slightly below the stimulus and it moved with the participant as they experienced visual self-motion. To keep the visual input identical across both tasks, the target rectangle from the prediction task, while irrelevant for the speed estimation task itself, was present in this task as well. Overall, participants performed between 30 and 37 trials in 18 staircases (two start values, three speeds and three motion profiles) for a total of between 540 and 666 trials.

Before proceeding to the main tasks, participants completed a training session. This training session consisted of one PEST of reduced length (between 20 and 27 trials) that started 30% above the speed of the ball (3 m/s). Participants needed to achieve a final step size of below 0.3; otherwise, they were asked to repeat the training. If they failed the training a second time, we excluded them from the analysis and all their data collected thus far were destroyed. The participants did not experience visually simulated self-motion in this training. This task–including the training–took about 40 minutes to complete.

Participants could choose to receive the instructions as PDF (can be downloaded from GitHub: https://github.com/b-jorges/Predicting-while-Moving/blob/main/Instructions%20Predicting%20while%20moving.pdf) or watch a video (which can be viewed on YouTube: https://youtu.be/qHTWVyjn0QI and https://youtu.be/JyOZ-duRGmU, respectively).

## Modelling the predictions

To obtain specific predictions corresponding to each hypothesis, we built models of the underlying perceptual processes for both the prediction and the speed estimation task. The instantiation of the model for the prediction task can be found here (on GitHub: https://github.com/b-jorges/Predicting-while-Moving/blob/main/Analysis%20Prediction.R), and the instantiation of the speed estimation model can be found here (on GitHub: https://github.com/b-jorges/Predicting-while-Moving/blob/main/Analysis%20Speed%20Estimation.R). The implementation of the model that relates performance in both tasks can be found here (on GitHub: https://github.com/b-jorges/Predicting-while-Moving/blob/main/Predictions%20Correlations.R). While a detailed discussion of these models can be found in Appendix A in S1 Appendix, the most important assumptions are the following. The first assumptions reflect our hypotheses:

- The speed of the ball is overestimated by 20% of the presented self-motion speed when observer and ball move in opposite directions [37], with a between-participant standard deviation of 30%. In the prediction task, this overestimation of speed should lead to an underestimation of the time it takes for the ball to travel the occluded distance. No biases are assumed for the Same Directions and Observer Static motion profiles. This assumption reflects Hypothesis 2a.

- While we previously did not find evidence for an impact of visual self-motion on precision [37], we believe that the higher self-motion speeds in this study might enable us to uncover small effects that were not apparent at lower self-motion speeds. The variability in perceived speed is 20% higher for the Opposite Directions motion profile, with a between-participant standard deviation of 30%. No differences in variability are assumed for the Opposite Directions and Static motion profile. This assumption reflects Hypothesis 2b.

- The same effects of self-motion on accuracy and precision are also at play in the prediction task. This assumption reflects Hypotheses 1a and 1b.

- Participants display the same biases in perceived object speed in response to self-motion in the opposite direction of the ball in both the speed estimation and the prediction task. Similarly, variability is impacted equally in both tasks. This assumption reflects Hypotheses 3a and 3b.

We further need to make several assumptions about how the participants process the stimulus independently of the presence of visual self-motion:

- We neglect how exactly participants recover physical speed from angular speed and the perceived depth but to acknowledge the added complexity, we assume a Weber Fraction of 10% for the estimation of the ball speed, which is slightly higher than is generally reported for less complex speed discrimination tasks [40], with a between-participant standard deviation of 1.5%.

- *Prediction task only*: We assume that the computations executed by the visual system are approximated accurately with the physical equation for distance from speed and time (d = v*t), such that the extrapolated time ($t_{extrapolated}$) can be estimated from the distance between the point of disappearance and the target ($d_{perceived}$) and the perceived speed of the ball ($v_{perceived}$):

$$t_{extrapolated} = d_{perceived} / v_{perceived} \tag{2}$$

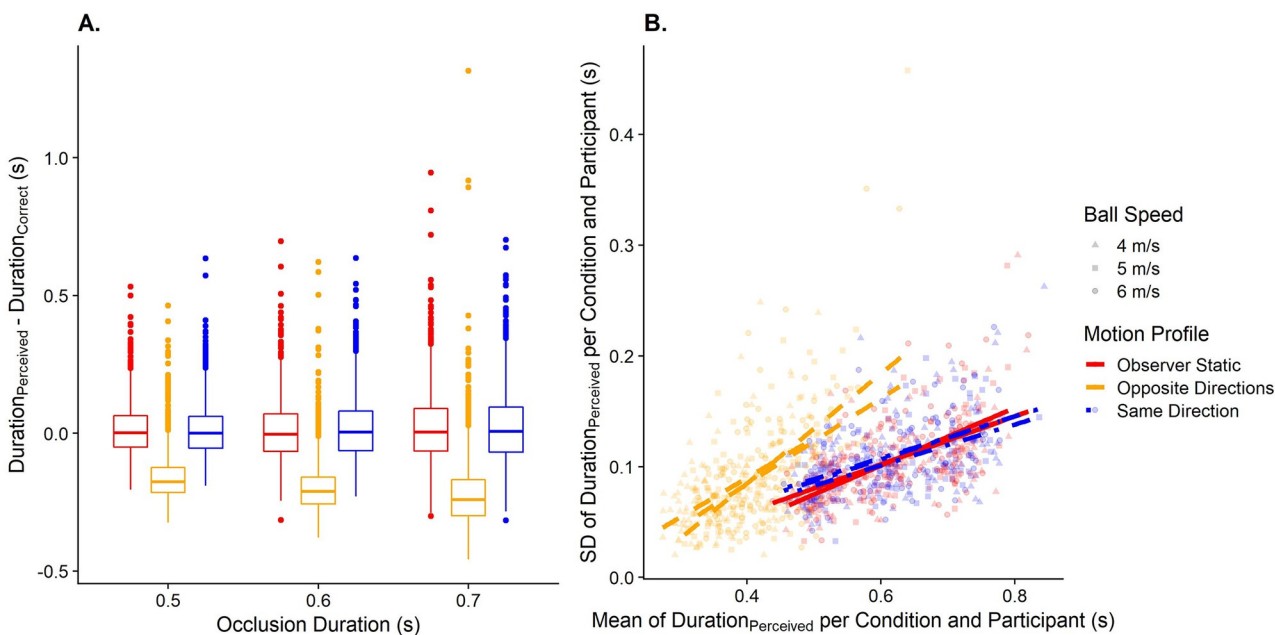

**Fig 3.** A. Predicted data for the timing error in the prediction task, divided up by occlusion durations (x axis) and motion profile (color-coded; left-most: "Observer Static"; in the middle: "Opposite Directions"; right-most "Same Directions"). The dashed horizontal line at y = 0 indicates perfect performance. B. Predicted data for variability in the prediction task. The y axis displays the standard deviation of the extrapolated duration per condition and participant, while the x axis corresponds to the mean of the extrapolated duration per condition and participant. The motion profile is coded with different colors and line types (red and continuous for "Observer Static", yellow and dashed for "Opposite Directions" and blue and dashed-and-dotted for "Same Direction"). The lines are regression lines for their respective condition.

- *Prediction task only*: The distance between the point of disappearance of the ball and the target rectangle (that is, the occluded distance) is estimated with a Weber Fraction of 5%, as reported in the literature [41]. We further assume that this distance is estimated accurately or, if biases in depth perception impact accuracy, that these biases also impact the perceived speed of the ball in such a way that these biases cancel out. Out of these two scenarios, we believe the latter is more likely, as it is quite established in the literature that depth is under-estimated in VR [42], and an underestimation of depth would lead to an underestimation of the physical distance between the point of disappearance and the target rectangle. However, this bias in depth perception should also lead the observer to underestimate the physical speed of the ball in the same way, causing both biases to cancel each other out. Between-participant variability is neglected here.

**Prediction.** For the prediction task, under the above assumptions, participants were expected to respond between 0.12 and 0.2s earlier during the Opposite Direction motion profile than during the Static motion profile (Fig 3A). Our model further predicted that visual self-motion during motion observation would lead to higher variability in responses. Measuring the relation between self-motion and variability is not straight-forward because self-motion should cause an underestimation of the occlusion duration. Noise should be proportional to the mean length of the extrapolated interval. A shorter predicted interval should thus in turn be related to lower variability (in absolute terms) even if self-motion has no direct effect on precision. Fig 3B illustrates the expected relationship between biases in prediction, the motion profile, and variability in responses.

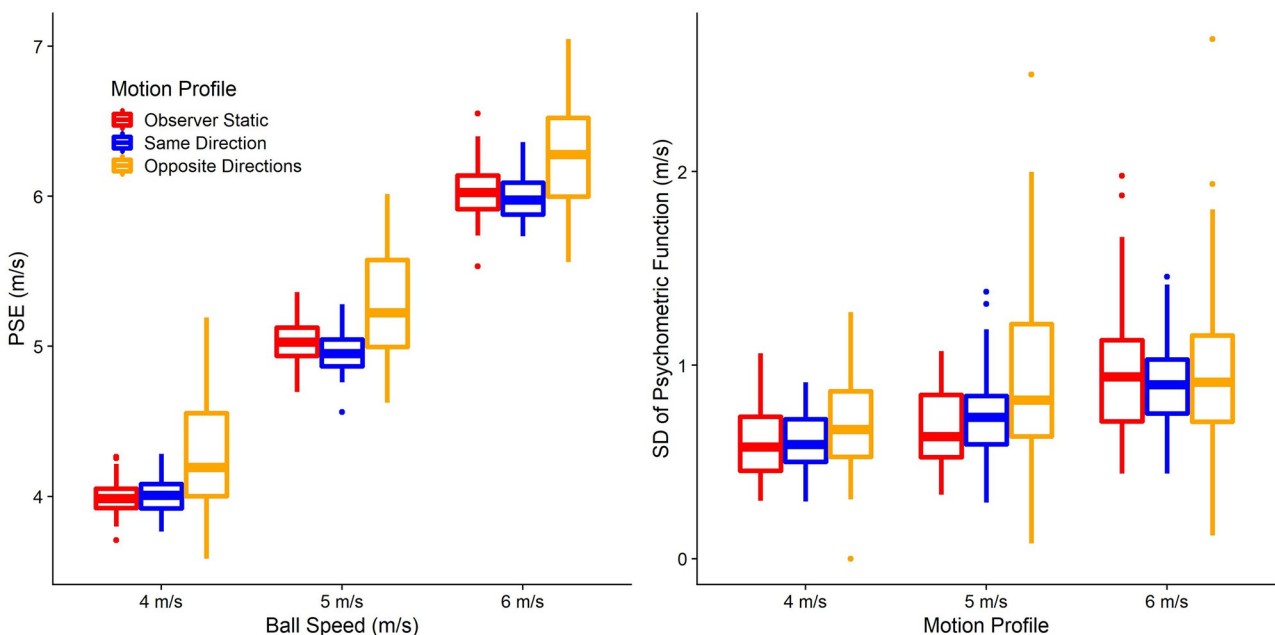

**Fig 4.** A. Predicted PSEs (y axis) for each ball speed (x axis) and motion profile (color-coded; left-most: "Observer Static"; in the middle: "Opposite Directions"; right-most "Same Directions"). B. As A. but for the predicted JNDs.

**Speed estimation.**   Here, we expected to replicate the findings from our previous study [37]: There, we found that participants largely estimated speed with the same degree of accuracy when they were static as when they were moving in the same direction as the target. In line with these results, visually simulated self-motion in the opposite direction to the ball should lead to an overestimation of ball speed (Hypothesis 2a; see Fig 4A). Since we used a higher self-motion speed than in our previous study, we also expected that precision would be lower for visual self-motion in the opposite direction to the ball (Hypothesis 2b, see Fig 4B).

**A link between speed estimation and predicted time-to-contact.**   We further expected the errors observed in the prediction task in response to self-motion to correlate with the errors in the speed estimation task in response to self-motion, indicating that performance in speed perception translated to errors in predicted time-to-contact, both in terms of accuracy (Fig 5A) and precision (Fig 5B).

## Data analysis

We first performed an outlier analysis. For the prediction task, we excluded all trials where the response timing was more than three times the occlusion duration (328 out of 10,920 trials, 3%), which indicated that the participant had not paid attention and missed the trial. For the speed estimation task, we excluded participants where more than 20% of presented ball cloud speeds were at the limits we set for the staircase (one third of the speed of the single ball and three times the speed of the single ball; 32 out of 360 staircases, 8.9% of all staircases). For all analyses related to precision, we further excluded all conditions where we obtained a standard deviation of 0.01 or lower (35 conditions in the prediction task, 1.83% of all remaining conditions; 1 staircase for the speed estimation task). According to our simulations, this should occur very rarely, and taking the log of such low values, as we do for the precision analyses to counteract the expected skew in these distributions, would lead to extremely small numbers

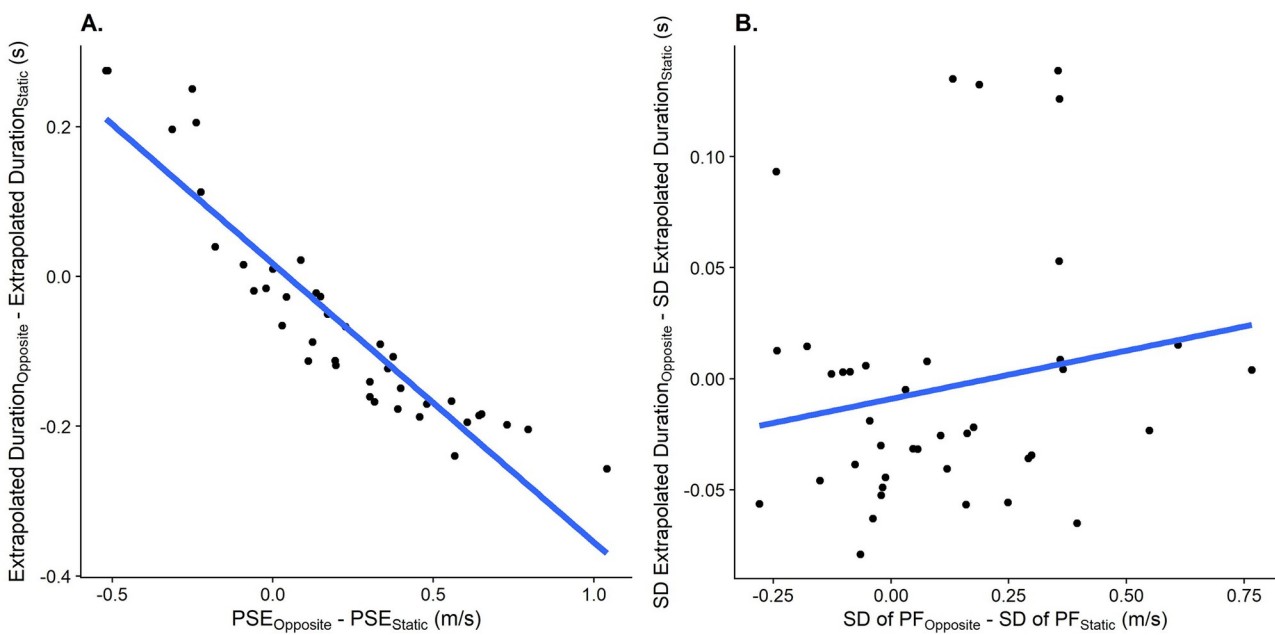

**Fig 5.** A. Relationship between the difference in PSEs between the Opposite Directions motion profile and the Observer Static motion profile in the speed estimation task (x axis) and the difference in predicted durations between these motion profiles (y axis). One data point corresponds to one participant. B. As A., but for the relation between the JND differences in the speed estimation task between the "Opposite Directions" motion profile and the "Observer Static" motion profile and the differences in standard deviations between these motion profiles.

that could bias results unduly. We also removed all trials where the head rotation was outside of the permitted range (+- 2.5°) for half or more of the recorded bins. (2 trials out of the remaining 10,592 trials in the prediction task, 0.02%; 45 trials out of the remaining 22,025 trials in the speed estimation task, 0.2%)

Unless noted otherwise, we computed bootstrapped 95% confidence intervals as implemented in the confint() function for base R [43] to determine statistical significance.

**Prediction.** To test Hypotheses 1a regarding accuracy, we used Linear Mixed Modelling as implemented in the lme4 package [44] for R. The corresponding script can be found here (on GitHub: https://github.com/b-jorges/Predicting-while-Moving/blob/main/Analysis%20Prediction.R). We fitted a model with the temporal error as dependent variable, the motion profile ("Observer Static", "Same Direction" and "Opposite Directions") as fixed effect, and random intercepts and random slopes for the speed of the ball per participant, as well as random intercepts for the occlusion duration as random effects. In Wilkinson & Rogers notation (1973), this model reads as follows:

$$Error \sim Motion\ Profile + (Speed_{Ball}\ |\ Participant)\ + (1|Occlusion\ Duration) \qquad (3)$$

We expected the regression coefficient corresponding to the motion profile "Opposite Directions" to be negative and significantly different from zero.

For Hypothesis 1b regarding precision, we needed to take into account one possible confound: differences in timing accuracy can impact variability: overestimating the time it takes the ball to hit the rectangle could be connected to a higher variability, while underestimating the time could lead to lower variability. For this reason, we first computed the means and standard deviations of extrapolated durations for each condition and participant. We then fitted a test model with the log standard deviations as dependent variable, the mean timing error and

the motion profile as fixed effects, and random intercepts as well as random slopes for ball speeds per participant and random intercepts for the occlusion durations as random effects:

$$\log(SD\ of\ Extrapolated\ Time) \sim Mean\ of\ Extrapolated\ Time + Motion\ Profile +$$
$$(Speed_{Ball} \mid Participant) + (1 \mid Occlusion\ Duration) \tag{4}$$

We further fit a null model without the motion profile as fixed effect:

$$\log(SD\ of\ Extrapolated\ Time) \sim Mean\ of\ Extrapolated\ Time +$$
$$(Speed_{Ball} \mid Participant) + (1 \mid Occlusion\ Duration) \tag{5}$$

We then compared both models by means of a Likelihood Ratio Test to determine whether the motion profile explains significantly more variability than the test model which already took into account biases in extrapolated time. We do not interpret the regression coefficients as means and standard deviations are likely to be correlated, which can lead to biased regression coefficients.

**Speed estimation.**   To test Hypotheses 2a and 2b (script can be found here, on GitHub: https://github.com/b-jorges/Predicting-while-Moving/blob/main/Analysis%20Speed% 20Estimation.R) regarding speed estimation, we first used the R package quickpsy [45] to fit psychometric functions to the speed estimation data, separately for each participant, speed and motion profile. Quickpsy fits cumulative Gaussian functions to the data by direct likelihood maximization. The means of the cumulative Gaussians correspond to the Points of Subjective Equality (PSEs) and their standard deviations correspond to the 84.1% Just Noticeable Differences (JNDs).

To assess whether the motion profile biased the perceived speed significantly, we fitted a Linear Mixed Model with the PSEs as dependent variable, the self-motion profile as fixed effect, and random intercepts and random slopes for the ball speed per participant as random effects:

$$PSE \sim Motion\ Profile + (Speed_{Ball} \mid Participant) \tag{6}$$

We expected that the regression coefficient for the motion profile "Opposite Directions" would be positive and significantly different from zero, indicating that speed was overestimated when observer and target move in opposite directions as compared to when the observer was static.

Regarding precision, the same considerations apply as for the prediction task: in addition to a direct effect of the self-motion profile, biases elicited by the different self-motion profiles can impact precision. For this reason, we used a model comparison-based approach similar to the one used above. Separately for the "Same Direction" and "Opposite Directions" motion profiles, we first fitted a test model that contained the log JNDs as dependent variable, the self-motion profile and the PSEs as fixed effects, and random intercepts as well as random slopes for ball speed per participant as random effects.

$$\log(JND) \sim Motion\ Profile + PSE + (Speed_{Ball} \mid Participant) \tag{7}$$

We also fitted a null model without the motion profile as fixed effect:

$$\log(JND) \sim PSE + (Speed_{Ball} \mid Participant) \tag{8}$$

Finally, we compared both models with a Likelihood Ratio Test and we expected the test model (Eq 7) to be a significantly better fit than the null model (Eq 8). As for the prediction task, we do not interpret the regression coefficients obtained in this analysis.

**A link between speed estimation and prediction.** To test Hypotheses 3a and 3b (script can be found here, on GitHub: https://github.com/b-jorges/Predicting-while-Moving/blob/main/Analysis%20Correlation.R), we first prepared the prediction data by computing means and standard deviations of the extrapolated time for each participant. We then calculated the difference in performance (mean and standard deviations for the prediction task and PSEs and JNDs for the speed estimation task) between the "Opposite Directions" motion profile and the "Observer Static" motion profile for both tasks for each participant.

For accuracy, we then determined to what extent PSE differences between the Opposite Direction motion profile and the Observer Static motion profile obtained in the speed estimation task predicted the mean extrapolated time in the prediction task. For this purpose, we fitted a Linear Model with the difference in mean motion extrapolation errors between the motion profiles as the dependent variable and the difference in PSEs between the motion profiles as the independent variable:

$$
\begin{aligned}
& Mean\ Difference\ in\ Extrapolated\ Time - to - Contact \\
& (Opposite - Static) \sim PSE\ Difference(Opposite - Static)
\end{aligned}
\tag{9}
$$

We expected that the regression coefficient for the fixed effect "PSE Difference (Opposite–Static)" would be negative and significantly different from zero, indicating that a stronger effect of self-motion on PSEs was linked to a larger effect of self-motion on the estimated time-to-contact.

For precision, the same complication as for Hypothesis 1b applies: A correlation between the effect of visual self-motion on the precision of speed estimation and on the precision of the predicted times-to-contact could be due to biases introduced by visual self-motion. If visual self-motion in the opposite direction, for example, leads to too-early responses, the extrapolated intervals become shorter. A shorter interval, in turn, would lead to higher precision. Therefore, to test whether the difference in precision observed in the speed estimation task was significantly related to the variability in the prediction task even after accounting for biases, we needed to determine whether the effect of visual self-motion on JNDs predicted any variability beyond the variability that was already explained by the bias in motion extrapolation. To test this hypothesis, we first fitted a test model with the variability difference between the "Opposite Directions" motion profile and the "Observer Static" motion profile in the prediction task as the dependent variable and the mean difference between these motion profiles and the difference in JNDs in the speed estimation task as the independent variables (as a measure of bias introduced by visual self-motion):

$$
\begin{aligned}
& Variability\ Difference\ in\ Extrapolated\ Time \\
& (Opposite - Static) \sim Mean\ Difference\ in\ Extrapolated\ Time \\
& (Opposite - Static) + JND\ Difference(Opposite - Static)
\end{aligned}
\tag{10}
$$

We also fitted a null model without the JND difference as independent variable:

$$
\begin{aligned}
& Variability\ Difference\ in\ Extrapolated\ Time \\
& (Opposite - Static) \sim Mean\ Difference\ in\ Extrapolated\ Time(Opposite - Static)
\end{aligned}
\tag{11}
$$

Then, we used a Likelihood Ratio Test to determine whether the test model (with the JND difference as fixed effect) was significantly better than the null model. We expected the test model (Eq 10) to be a significantly better fit than the null model (Eq 11). As above, the regression coefficient was not interpreted.

**Model fitting.** These analyses only serve to demonstrate that performance in both tasks is related, but they don't provide insight into how strongly they are related. We will therefore use the models outlined in the section "Modelling the Predictions", and described more in detail in Appendix A in S1 Appendix, to fit parameters that capture biases and precision differences in perceived speed due to self-motion (which we had set to 20% on average to model our predictions and conduct the power analyses) in both tasks separately.

We used a two-step approach to fit these parameters: since we expected biases to affect variability in performance but did not expect variability to affect biases, we first set the variability parameter (capturing the impact of self-motion on the variability of perceived speed) to zero and fitted the accuracy parameter (capturing the impact of self-motion on mean perceived speed). To do so, we minimized the root median squared error between the observed mean difference in timing errors between baseline and the self-motion condition and the difference in timing error in the simulated dataset across all speeds and occlusion durations. We performed this optimization by using the Brent method [46] as implemented in the optimize function in base R. In the second step, we set the accuracy parameter for each participant to the one fitted in the first step and fit the precision parameter. Here, we minimized the root median squared error between the difference between the standard deviation of the observed difference in timing errors in the baseline condition and the self-motion condition and the respective simulated values. We used the same approach to obtain these parameters for the speed estimation task as well but used the observed and simulated PSEs (for accuracy) and JNDs (for precision).

Once we obtained one accuracy parameter and one precision parameter for each participant and task, we performed a simple linear regression between the parameters fitted for the prediction task and the speed estimation task (separately for accuracy, see Eq 11, and precision, see Eq 13) to determine to what extent performance in one task was indicative of performance in the other:

$$Bias_{Prediction} \sim Bias_{Speed\ Estimation} \qquad (12)$$

$$Precision\ Difference_{Prediction} \sim Precision\ Difference_{Speed\ Estmation} \qquad (13)$$

Since these parameters were scaled the same way for both tasks (the effect of self-motion on accuracy/precision as a fraction of presented self-motion speed), we expected regression coefficients of around 1 in both analyses. A value of above 1 would mean that the effect of self-motion was stronger in the prediction task than in the speed estimation task, and vice-versa. To test this prediction, we computed 95% confidence intervals which we expect to contain a value of 1.

**An effect of visual self-motion in the same direction as the ball.** While our earlier results [37] suggested that visual self-motion in the same direction as the observer should not have any effect on perceived speed, we performed all analyses outlined in this section equivalently for the "Same Direction" motion profile as well.

## Power analysis

Since power for complex hierarchical designs cannot be computed analytically, we used Monte Carlo simulations (as per [47]) to determine power for all statistical tests outlined in the previous section: we used our models for the prediction task and the speed estimation task to first simulate full datasets. Then, we performed the analyses detailed above over each of these simulated datasets and determined the results for each combination of number of participants and number of trials. To keep the computational cost manageable, we used the faster, but more

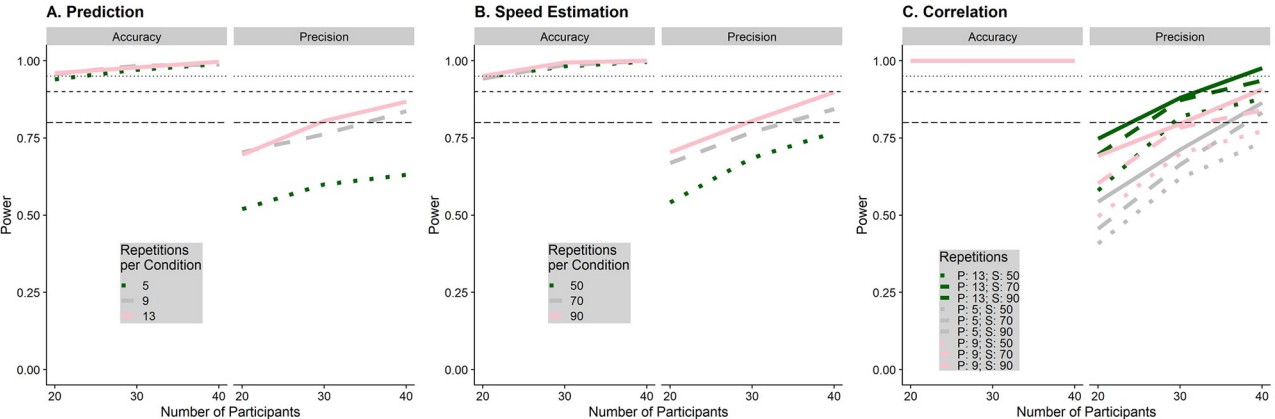

**Fig 6.** Simulated power for the prediction task (A), the speed estimation task (B) and the correlation between performance in speed estimation and speed prediction (C), separately for the statistical tests referring to biases (accuracy) and variability (precision). The number of participants for which we simulated power is on the x axis, while the number of trials for each task is coded with different shades of green and line types. The horizontal lines indicate a power level of 0.8, 0.9 and 0.95 respectively.

bias-prone Satterthwaite approximation, as implemented in the lmerTest package [48] for R, to assess statistical significance rather than bootstrapped confidence intervals. The script used for the power analyses can be found here (on GitHub: https://github.com/b-jorges/Predicting-while-Moving/blob/main/Power%20Analysis.R).

We repeated this process 250 times for all combinations of 20, 30 and 40 participants, 5, 9 and 13 repetitions per condition in the prediction task, and 20 to 27, 30 to 37, and 40 to 47 trials per pest, which makes for an average of 50, 70 and 90 trials per condition, respectively, for the speed estimation task. The results are shown in Fig 6. For the precision in the prediction task, using 9 repetitions per condition appears to add a considerable amount more power than using only 5 repetitions, while the added benefit of another 4 repetitions for a total of 13 is small. However, the prediction task is very quick to do, taking only around 10 minutes even with 13 repetitions per condition. Similarly, 70 trials per condition increases the power to detect an effect on precision significantly more than using only 50 trials, while the added benefit of 90 trials is marginal. Since the speed estimation task takes much longer and is more fatiguing than the prediction task, we judge this marginal increase in power to be not worth the additional time spent by the participant. We thus opt for a combination of 40 participants, 13 repetitions per condition in the prediction task, and 70 trials per condition in the speed estimation task, which allows us to achieve a power of at least 0.85 for all statistical tests.

We also used our power analyses to determine that all of our statistical analyses led to an expected false positive rate of 0.05 in absence of a true effect.

## Results

### Statistical analysis

**Hypothesis 1a: Effect of visual self-motion on accuracy in motion extrapolation.** The Linear Mixed Model (see Eq 3) fitted to assess Hypothesis 1a had an intercept of 0.09 (95% CI = [0.02; 0.16]). Participants pressed the button significantly earlier in the Motion Profile: Opposite Directions (by 0.055s, 95% CI = [0.041s;0.07s]) than in the Static condition, indicating a relative underestimation of time-to-contact in line with our hypothesis 1a. They further

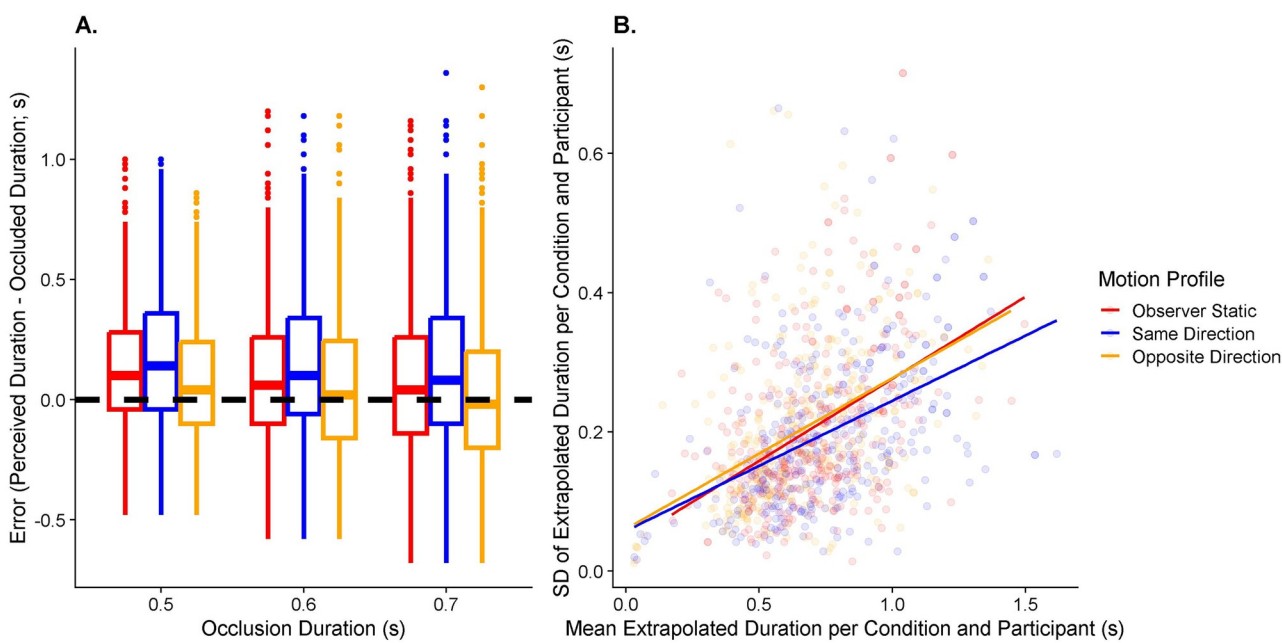

**Fig 7.** A. Boxplots of the timing error (y axis) separately for occlusion duration (x axis) and motion profile (color-coded). The dashed horizontal line at y = 0 indicates perfect performance; positive numbers mean that the participant pressed the button too late, and negative numbers mean that the participant pressed the button too early. B. Variability in timing responses (y axis) as a function of bias (x axis), separated by motion profile (color-coded). The lines correspond to a linear regression of the variability on the bias across all participants and target speeds, only separated by motion profile. Data in these plots are restricted to the three main occlusion durations (0.5s, 0.6s, and 0.7s).

pressed the button significantly later in the Motion Profile: Same Direction (by 0.047s, 95% CI = [0.032s;0.064s]). These results are illustrated in Fig 7A.

**Hypothesis 1b: Effect of visual self-motion on variability in motion extrapolation.** The likelihood ratio test used to compare the respective models (Eqs 4 and 5) for the Motion Profile: Opposite Directions yielded a non-significant result, i.e., contrary to our hypothesis 1b, precision was not significantly different in the Opposite Directions motion profile than when the observer was static. However, the test model (Eq 4) was significantly better than the null model (Eq 5) for the Motion Profile: Same Direction (p = 0.006), which taken together with a negative regression coefficient in the test model means that variability was–surprisingly–*lower* when participants experienced self-motion in the same direction as the observer. See Fig 7B for an illustration of the results. Please note that a simple boxplot is insufficient to illustrate the effect of self-motion on precision because differences in accuracy due to self-motion are likely to impact precision as well and in a boxplot, it is impossible to distinguish between the direct effect of self-motion on precision and the indirect effect (mediated by changes in accuracy).

**Hypothesis 2a: Effect of visual self-motion on accuracy in speed perception.** The model used to assess biases in perceived speed in function of the motion profile (Eq 6) had an intercept of 3.3 m/s (95% CI = [2.92 m/s;3.7 m/s]). Visual self-motion in the opposite direction led to an overestimation of target speed by 1.08 m/s (95% CI = [0.86 m/s;1.32 m/s]) relative to a static observer, i.e., about 30% of the mean self-motion speed (of 3.6 m/s) was–on average–wrongly attributed to object motion rather than self-motion. No significant difference was detected between the Static and Same Direction motion profiles. See Fig 8A.

**Hypothesis 2b: Effect of visual self-motion on variability in speed perception.** When comparing the models to assess precision differences using Likelihood Ratio Tests, we found

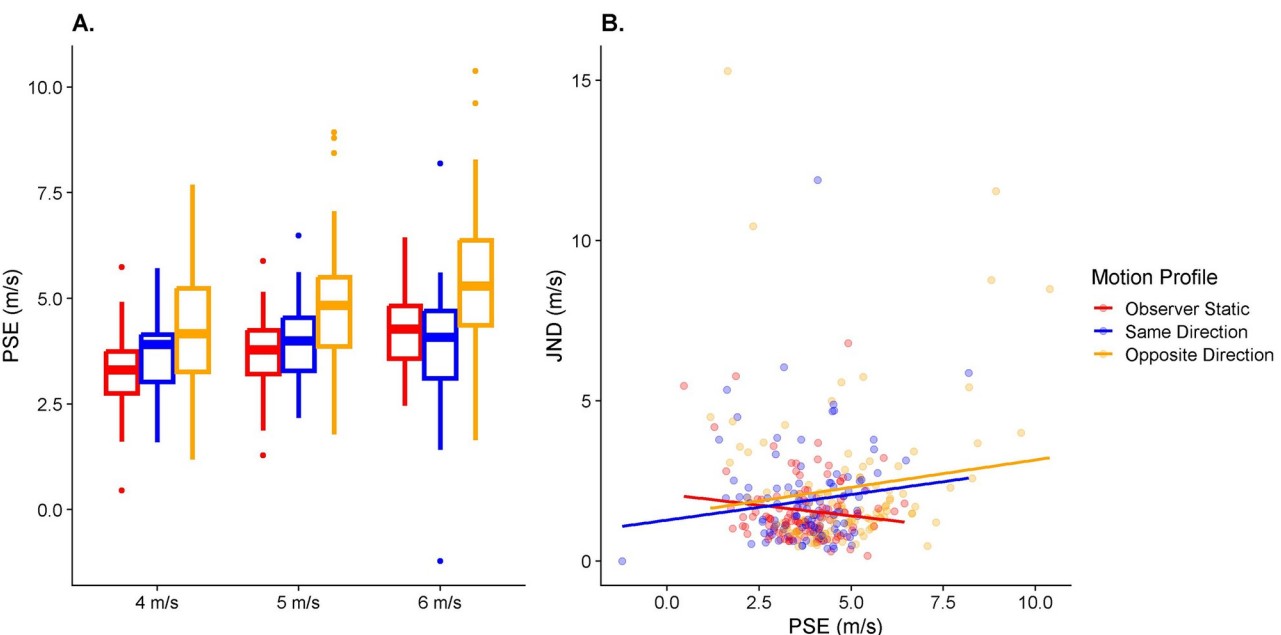

**Fig 8.** A. Boxplots of the fitted PSEs (y axis) separately for target speed (x axis) and motion profile (color-coded). B. Fitted JNDs (y axis) as a function of PSEs (x axis), separated by motion profile (color-coded). The lines correspond to a linear regression of JNDs on PSEs across all participants and target speeds, only separated by motion profile.

that for both motion profiles, the test model (Eqs 7 and 10) was significantly better than the null model (Eqs 8 and 11; p = 0.007 for Motion Profile: Opposite Directions, and p = 0.038 for Motion Profile: Same Direction), indicating that self-motion had an effect on variability. When examining the regression coefficients, we found a positive association for both motion profiles, i.e., variability was significantly higher when the observer was moving either in the same or in the opposite direction of the target than when they were static, even after taking into account expected increases in variability due to the effect of self-motion on accuracy. Fig 8B illustrates this result.

**Hypothesis 3a: A Correlation between biases in speed judgements and biases in motion extrapolation?.** The linear regression model (Eq 13) fitted to assess whether biases in speed judgements were associated with corresponding biases in motion extrapolation in the Motion Profile: Opposite Directions had an intercept of -0.02 (95% CI = [-0.05;0.02]). The regression coefficient for the association between each participant's bias in speed perception due to self-motion and their bias in motion extrapolation due to self-motion was negative (-0.03, 95% CI = [-0.05;-0.01]), i.e., participants that displayed a larger bias in response to self-motion in the opposite direction of the target also tended to press the button earlier and vice-versa. No such correlation was evident for the Motion Profile: Same Direction. See Fig 9A and 9C.

**Hypothesis 3b: A correlation between precision in speed judgements and precision in motion extrapolation?.** For the Motion Profile: Opposite Directions, we found that the test model (Eq 10) was better (p = 0.03) than the null model (Eq 11) and the regression coefficient was positive, i.e., higher JNDs in the speed estimation task were related to higher variability in the motion extrapolation task. For visual self-motion in the same direction as the target, the test model (Eq 10) was not significantly better than the null model (Eq 11), indicating no evidence for a relationship between variability in speed judgements and variability in motion extrapolation. See Fig 9B and 9D.

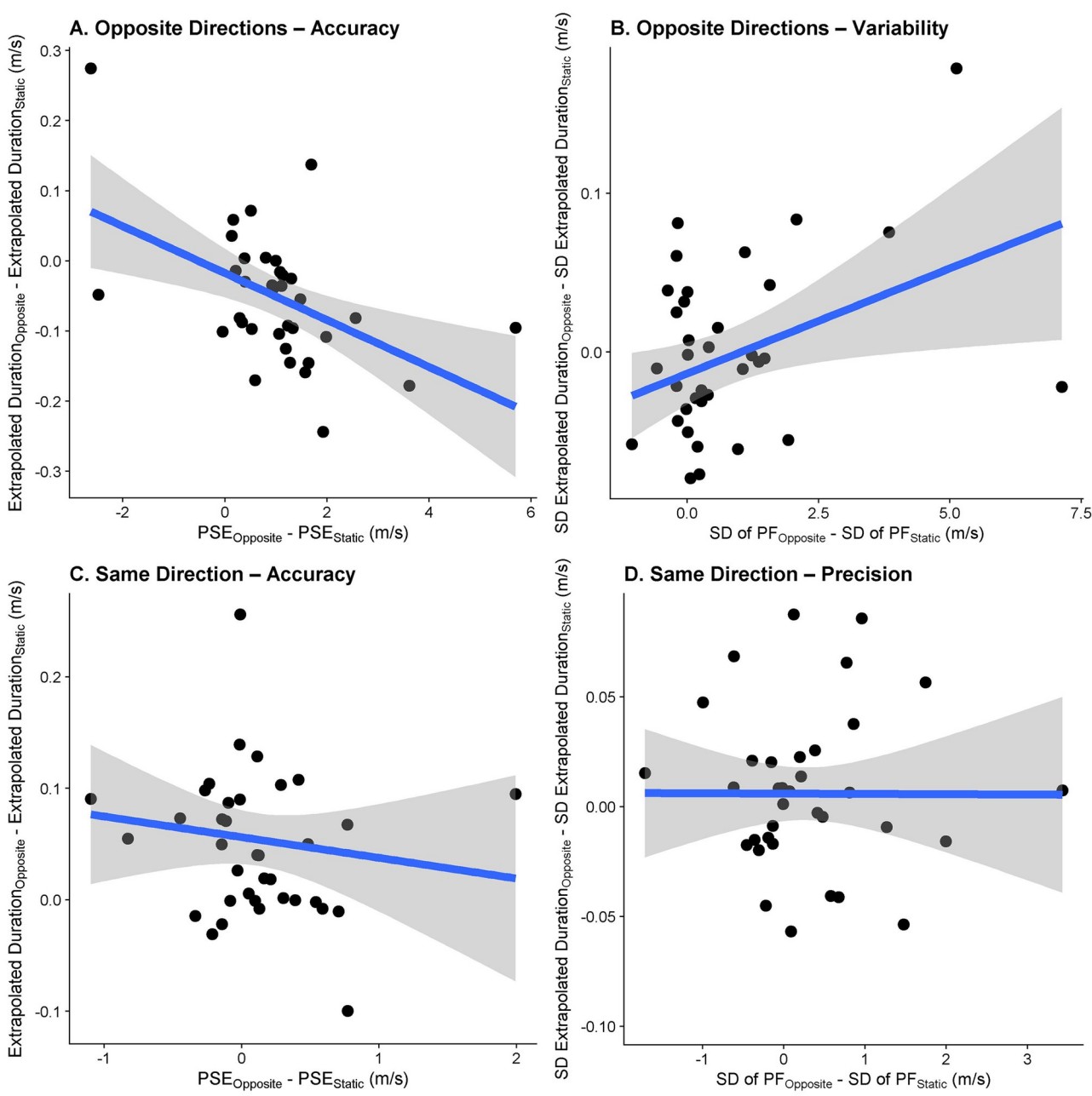

**Fig 9.** The relationships between biases in speed perception due to visual self-motion (x axis) and biases in motion extrapolation (y axis) due to visual self-motion for the motion profiles Opposite Directions (A) and Same Direction (C), as well as the relationships between the effect of visual self-motion on variability in motion extrapolation and the effect of visual self-motion the variability in perceived object speed for the Motion Profile: Opposite Directions(B) and the Motion Profile: Same Direction (D). Each data point corresponds to one participant. Each data point corresponds to one participant.

## Modelling

In the above statistical analysis, biases could misleadingly scale the variability of the responses. The following method circumvents this potential confound: we fitted the models (see Appendix A in S1 Appendix) that we used to make our predictions and generate the power analyses

to our data in order to obtain less biased estimates of the effect of self-motion on precision. In the process, we also fitted parameters relating to the effect of self-motion on accuracy, which we expected to be in agreement with the results of the statistical analysis in the previous section. You can find the script on GitHub (https://github.com/b-jorges/Predicting-while-Moving/blob/main/Fit%20Models.R). Specifically, we used two-step fitting process in which we first fitted a parameter capturing the size of the effect of self-motion on accuracy (which we assumed not to be affected by any differences in variability induced by self-motion) as a fraction of the presented self-motion speed (with -100% of the self-motion speed as lower bound and +100% as upper bound), while setting the parameter capturing the effect of self-motion on variability to 0. In the second step, we set the accuracy parameter to the value obtained in the first step of the fitting process and let the precision parameter vary. We did this separately for each participant, task (prediction versus speed estimation) and self-motion direction (Same versus Opposite directions).

Since our model incorporated a certain amount of noise, on each iteration of the optimization process we generated 50 (for the prediction task) and 25 (for the speed estimation task) datasets. For the prediction task, we computed summary statistics (mean errors as a measure of accuracy and standard deviations per condition as a measure of precision) for each of these simulated data sets and the relevant subset of the observed data. We then took the difference between the self-motion condition (Opposite or Same) and the Static condition (regarding means and standard deviations) as a measure of simulated and observed impact of self-motion on accuracy and precision and computed the root median square error between the simulated and observed values. In the first step, we minimized the median RMSE in simulated and observed means across all 50 datasets to obtain the accuracy parameter that produced the best fit (using the optimize() function from base R which relies on a combination of the golden section search and successive parabolic interpolation), and in the second step we minimized the median RMSE in simulated and observed standard deviations to obtain the precision parameter. For the speed estimation task, we followed a similar procedure, except that rather than means and standard deviations we used fitted PSEs and JNDs as measure of accuracy and precision, respectively.

This way, we obtained one parameter per participant and

- Task (Prediction versus Speed Estimation)

- Motion profile (Same versus Opposite directions)

- Accuracy and Precision

Please note that this parameter is coded in such a way that a positive value means (for both tasks) that the underlying speed was overestimated, and a negative value means that the underlying speed was underestimated. The distributions of values for each of these 8 domains are depicted above (prediction) and to the right (speed estimation) of the scatterplots in Fig 10. We also correlated the values obtained for the Prediction task and the values obtained for the Speed Estimation task (see scatterplots in Fig 10).

We then assessed whether the effects of self-motion were significantly different from zero for any of the domains using an intercept-only linear regression (equivalent to a one-sample t-test), the results of which you can find in Table 2. We also fitted linear regressions to gauge the correlation between the effect of self-motion in both tasks (see Table 3). Overall, when analyzing the fitted parameters, we found a pattern that was largely similar to what we found in the raw data: self-motion in the opposite direction increased the perceived speed underlying both in the prediction and in the speed estimation task and both were correlated significantly (Fig 10A). Precision was only decreased in the speed estimation task while no effect was detected in

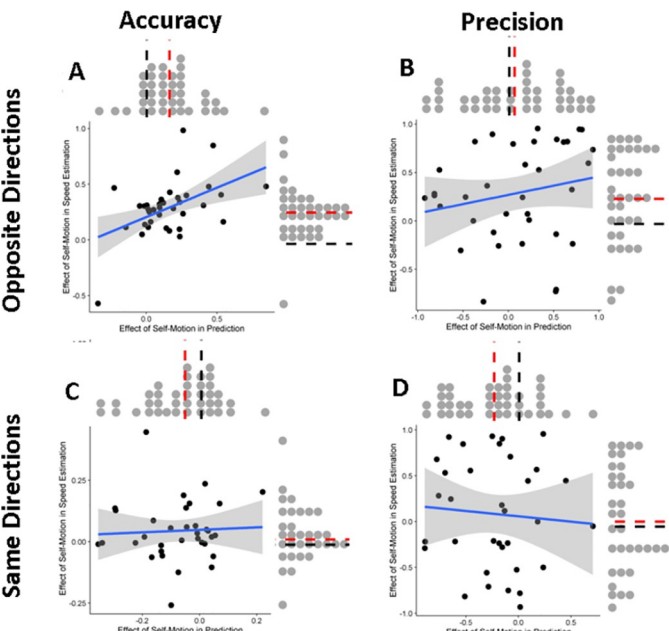

**Fig 10.** The main panels show the regression lines (in blue) between the fitted effect of self-motion on accuracy (A, C) and precision (B, D) for Opposite Directions (A, B) and Same Direction (C, D) motion profiles in the prediction task (x axis) and the speed estimation task (y axis), fitted using the geom_smooth() function from ggplot2 with the argument method = "lm" which also generates a confidence band (grey shaded area). The grey dots above the scatterplots indicate the distribution of fitted effects of self-motion in the prediction task, while the grey dots to the right of the scatterplots indicate the distribution of fitted effects of self-motion in the speed estimation task. We marked 0 with a black dashed line and the mean of the distribution with a red dashed line.

**Table 2. Results of the statistical tests (in terms of means and p values) performed over the fitted parameters capturing the effect of self-motion on different dependent variables.** We shaded the results in red when they are significantly different from zero and negative and blue when they are significantly different from zero and positive. For precision, values above zero signify an increase variability, i.e., a decrease in precision, and values below zero signify a decrease in variability, i.e., an increase in precision.

| | Mean | p value |
|---|---|---|
| **Prediction Task** | | |
| *Opposite Directions* | | |
| Accuracy | 0.16 | < 0.0001 |
| Precision | 0.06 | 0.49 |
| *Same Direction* | | |
| Accuracy | -0.06 | 0.02 |
| Precision | -0.24 | < 0.001 |
| **Speed Estimation** | | |
| *Opposite Directions* | | |
| Accuracy | 0.3 | < 0.0001 |
| Precision | 0.3 | < 0.001 |
| *Same Direction* | | |
| Accuracy | 0.04 | 0.04 |
| Precision | 0.08 | 0.4 |

**Table 3. Regression coefficients and p values for the correlations between the fitted parameters for the prediction task and the speed estimation task.** We shaded the results in red when they are significantly different from zero and negative and blue when they are significantly different from zero and positive.

|  | Regression Coefficient | p value |
|---|---|---|
| *Opposite Directions* |  |  |
| Accuracy | 0.43 | 0.001 |
| Precision | 0.13 | 0.47 |
| *Same Direction* |  |  |
| Accuracy | 0.06 | 0.8 |
| Precision | -0.05 | 0.67 |

the prediction task. The parameters were also not correlated significantly between both tasks (Fig 10B). For the Same Direction motion profile, the fitted parameters were slightly negative, indicating an underestimation of speed. This was not observed for the speed estimation task and the correlation between both was not significantly different from zero (Fig 10C). We further observed an increase in precision in the prediction task but no effect of self-motion in the speed estimation task and no correlation between prediction and speed estimation (Fig 10D).

## Discussion

Table 4 summarizes how the results relate to our three hypotheses, separately for the two motion profiles (Opposite Directions and Same Direction). Where the results of our two types of analyses (conventional statistics and modelling) disagree, we note such discrepancies.

### Opposing self and object motion directions

Overall, we found support for most of our hypotheses when observer and target moved in opposite directions: participants underestimated time-to-contact (Hypothesis 1a; mirroring other studies that showed biased time-to-contact estimates due to manipulations of perceived stimulus speed [31,32]), They also overestimated the target's speed (Hypothesis 2a), and participants who underestimated time-to-contact more tended to overestimate the speed of the target more (Hypothesis 3a). These results are therefore in line with a series of other studies reporting incomplete flow parsing in which at least part of the visual motion due to self-motion is attributed to the object [20,22–24,26,30,49–52]. Further, speed was estimated less precisely (Hypothesis 2b) and participants who displayed a stronger increase in variability in motion extrapolation were also less precise in their speed judgements (Hypothesis 3b). The

**Table 4. Overview over the results relating to all hypotheses.**

|  | Hypothesis 1a | Hypothesis 1b | Hypothesis 2a | Hypothesis 2b | Hypothesis 3a | Hypothesis 3b |
|---|---|---|---|---|---|---|
|  | *Self-motion biases motion extrapolation (Fig 7A)* | *Self-motion increases variability in motion extrapolation (Fig 7B)* | *Self-motion biases perceived speed (Fig 8A)* | *Self-motion increases variability in perceived speed (Fig 8B)* | *Biases in perceived speed correlate with biases in motion extrapolation (Fig 9A)* | *Variability increases in perceived speed correlated with variability increases in motion extrapolation (Fig 9B)* |
| Opposite Directions | Supported | Not supported | Supported | Supported | Supported | Supported (statistics) Not Supported (modelling) |
| Same Directions | Supported | Opposite supported | Not supported (statistics) Opposite supported (modelling) | Supported (statistics) Not supported (modelling) | Not supported | Not supported |

latter was only the case in the conventional statistical analysis, while no such relation rose to statistical significance when analyzing the fitted model parameters. It should be noted that the fitted model parameters are likely to give a more accurate picture of the underlying processes when it comes to precision because they are fitted before the expected changes in (absolute) standard deviations due to changes in accuracy come into play. The only hypothesis not supported was Hypothesis 1b, where we found no differences between visual self-motion and a static observer in terms of variability in motion extrapolation.

## Same self and object motion directions

When participants were moving in the same direction as the target, we found much more mixed and even surprising results: in line with our hypotheses, participants overestimated time-to-contact (Hypothesis 1a) and displayed increased variability in speed judgements (Hypothesis 2b) when analyzing the raw data (but not when analyzing the fitted model parameters). On the other hand, we found no difference between a static observer and visual self-motion in perceived speed (Hypothesis 2a) in the raw data, whereas a very small, barely significant increase of perceived speed was found in the fitted model parameters. Biases (Hypothesis 3a) or differences in precision (Hypothesis 3b) were not correlated significantly between motion extrapolation and speed perception. Finally, and surprisingly, we found that precision in motion extrapolation was *higher* during visual self-motion than for static observers (Hypothesis 1b).

## The asymmetry between the two motion profiles

As per our previous results [30], we expected to find our hypotheses supported when target and observer visually moved in opposite directions. This was largely the case here, with the notable exception of Hypothesis 1b, i.e., *self-motion would not lead to higher variability in time-to-contact estimates*, which surprisingly was not supported despite the presence of self-motion *and* the higher retinal speeds caused by self-motion. This suggests that there may be compensatory effects at play that allow participants to perform as precisely (or even more) as when static, even though any such compensatory effects seem unable to overcome the biases introduced by self-motion. One possibility is a Bayesian combination of sensory input with a regression-to-the-mean-like prior: over the course of the 278 trials in the prediction task, participants could build up an expectation of the average time it took the object to reach the target. Such an expectation would then have two effects: firstly, and perhaps most obviously, it would bias responses towards this average time across all conditions (target speeds, occlusion times, motion profiles). And in fact, while such a tendency would not overrule the expected effects of self-motion, it would cause what seems to be case that longer occlusion times would lead to somewhat earlier button presses relative to the correct time-to-contact and vice-versa–in line with such a regression to the mean. Secondly, integrating sensory information (i.e., the likelihood in a Bayesian framework) with such a hypothetical prior would increase the overall precision of the data. This, in turn, would lead to less of a difference in time to button press between the self-motion conditions than those predicted by our (simple, non-Bayesian) model described in Appendix A in S1 Appendix. Since our power analysis was based on such a simple, non-Bayesian model, our study may have been underpowered and unable to detect any remaining subtle differences in precision if the visual system did indeed integrate sensory information with such a regression-to-the-mean-like prior. The reason why the effects of such a prior might be stronger in the prediction task than in the speed estimation task (where we did find differences in precision between the motion profiles–see Fig 8B) is that in the prediction task, such priors could come into play twice: once when estimating the speed of the object

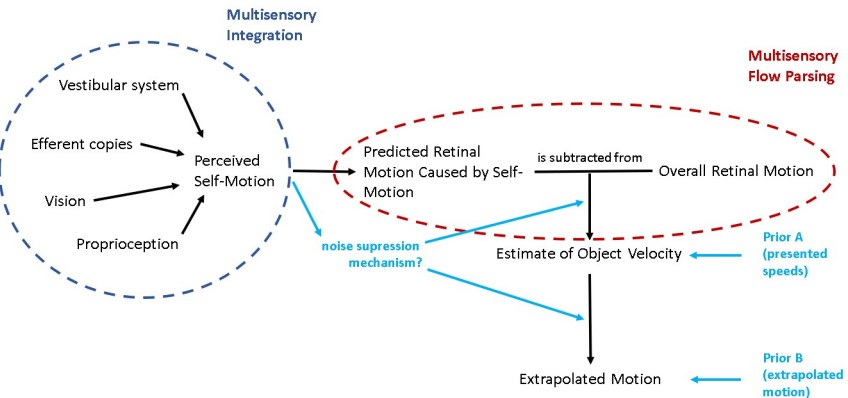

**Fig 11.** A schematic of our model of the data generating processes at play for motion prediction in the presence of self-motion, updated (in light blue) with the two places in which regression-to-the-mean-like priors could come into play (see "Asymmetry between the two motion profiles" section of the discussion), as well as with a potential noise suppression mechanism for predicting object motion during self-motion (see "Variability" section of the discussion).

(Prior A in Fig 11), and again when using this speed information to predict the future trajectory of the motion (Prior B in Fig 11). This is shown diagrammatically in an updated schematic of our model in Fig 11. If there were indeed two such priors at play, we would expect a stronger regression to the mean for motion extrapolation than for speed perception. However, our data do not allow us to evaluate this consequence of the updated model because the speed perception task was much harder than the prediction task. For the speed prediction task, participants needed to hold one motion in memory before making the judgement and they had to compare speeds between two different types of motion (a single sphere and a sphere cloud). A higher degree of difficulty is likely to result in a less reliable estimate, which in turn makes the speed estimation task much more susceptible to such a hypothetical prior.

## A perception-action dichotomy?

A further complication for our initial model is that the effects found for the Opposite Directions motion profile were not present in a roughly symmetrical fashion for the Same Directions motion profile in the speed estimation task (see Fig 8A), while results *were* symmetrical for the motion extrapolation task (see Fig 7A). A potential explanation for this asymmetry might be found in our self-motion profile and in the nature of the speed estimation task. To provide a more realistic experience, the participants' self-motion speed was ramped up over the first 50 ms of the 500 ms motion duration and slowed over the last 50 ms before coming to a halt, while the target moved at a constant speed throughout the whole 500 ms. Participants may have relied more on those transient parts of the motion profile which would have created transiently larger relative speeds between target and observer (see shaded areas in the "Same Direction" panel in Fig 2E). These higher speeds may increase the saliency of these parts of the trajectory (which has been shown, for example, in visual search [53]), i.e., participants' attention may have been drawn to these parts of the trial where the relative speed between observer and target was higher. While speed judgements over such short time periods are generally very unreliable [54], having two periods of higher relative speed (at the beginning and end of the motion) may have been sufficient to allow participants to use them instead of the longer steady state values. For the more action-oriented prediction task, participants may have instead integrated all the information available to them over the course of the trial, leading to biased results in both self-motion profiles. Given this possible explanation, different mechanisms may have

been at play for (1) the more perceptual speed judgements and (2) the more action-oriented time-to-contact estimates. This difference would then be an extension of the perception-action divide found in other domains such as the perception of slant [34], the reproduction of remembered locations [35] or kinaesthetic illusions [33], and would also be reminiscent of the vision-for-action versus vision-for-perception divide proposed by Milner and Goodale [55,56]. Such a division might explain why we were unable to detect a correlation in biases between our two tasks for the same directions motion profile. From an ecological perspective [57], this would make sense as speed judgements, other than the more intuitive, action-oriented and automatic time-to-contact judgements, require a higher degree of awareness and cognitive control.

## Variability

A final discrepancy between our predicted and observed results is that variability was actually *lower* for self-motion in the same direction as the observer in the prediction task whereas hypothesis 2b predicted the opposite. An initially appealing explanation is the large difference in retinal speeds between the motion profiles: they were vastly higher in the Opposite Directions condition (0.4–11.6˚/s) than for a static observer (22.2–34˚/s), and much lower than for the Same Direction condition (44.4–55.8˚/s). Higher retinal speeds should, according to Weber's Law, lead to higher variability. This then would be expected to be the case when target and observer moved in opposite directions. If Weber's Law were the only effect influencing variability, we would expect to see large differences between the three motion profiles. However, wherever we found any differences at all in variability between the conditions, they were extremely small (see Figs 7B and 8B). While this lack of variability is in line with results from our previous study [30], where we similarly found negligible variability differences in response to different retinal speeds and only minor differences in response to self-motion, this is a surprising finding both in terms of other studies investigating multisensory flow parsing [20] and in terms of the proposed theoretical background: mathematically, subtracting self-motion information (with its associated noise) from a global optic flow estimate (with its own associated noise) to obtain an estimate of object speed should lead to a vastly increased noise level compared to a scenario where the observer is static.

Two potential explanations come to mind: first, it might be that other sources of variability (e.g., related to the memory component or the transfer between two types of motion in the speed task, or the motor component, or even a lack of attention due to the boring nature of the experiment in both tasks) might dominate the overall variability in the measure, swamping out the expected changes. However, the pattern observed in the prediction task (lower variability for Same Direction and equal variability for Opposite Directions, see Fig 7B) suggests that, rather, there might be some kind of active compensatory mechanism at play that decreases variability in the presence of self-motion to below the levels of variability observed for a static observer. Such a mechanism would be useful as humans are more often than not in motion themselves when trying to interact with moving objects. While we can't draw any firm conclusions about any such mechanism, this represents a promising direction for future research.

## Flow parsing for lateral, forwards/backwards and rotational self-motion

Our results (with lateral self-motion) fit in well with studies that use either forwards/backwards (with radial optic flow) or rotational self-motion, which have generally found incomplete flow parsing when self-motion was only simulated visually ([11,18,20,58–60]). It is somewhat surprising how consistent these results are given that the geometrical solutions for flow parsing differ substantially between rotational, forwards/backwards, and lateral

self-motion: for rotational self-motion and object motion, the local difference in motion signals between object and background are sufficient to detect and/or judge object speed. For an accurate solution in the case of lateral or backwards/forwards self-motion and object motion, further computations are necessary, e.g., to recover the distance between object and observer. Also from an ecological perspective, one would expect differences between these motion profiles as they can occur under very different circumstances. Flow parsing in depth, i.e., in the case of forwards and backwards self-motion and object motion, is highly relevant and employed constantly in many daily situations (for example driving or biking or playing ball sports), while lateral and rotational flow parsing are useful on more rare occasions.

All changes with regards to the Registered Report Protocol are noted in Appendix B in S1 Appendix.

## Conclusions

Our initial model for object motion prediction when viewed during self-motion was based on the idea that incomplete (multisensory) flow parsing biases perceived speed [30] which we in turn expected to translate to biases in estimated time-to-contact. This model proved indeed largely accurate in predicting biases in time-to-contact estimation. However, the predicted changes in variability were not observed and no ad hoc extensions of the model could explain the full range of results.

It is particularly striking that the presence of self-motion can lead to a decrease in variability when the relative speed between target and observer is low without leading to an increase in variability when the relative speed between target and observer is high. Shedding light on this paradoxical finding by investigating a potential compensatory mechanism is what we would deem the most interesting and fruitful direction for future research. One possibility might be to investigate whether this observation holds up under more complete self-motion cues (e.g., in the presence of vestibular stimulation or even active locomotion on top of the visual cues presented in this study): Dokka et al. [20], for example, found a positive correlation between completeness of flow parsing and noise in object trajectory judgements in a multisensory paradigm. In contrast, a compensatory (potentially evolutionary) mechanism like the one we posit, should be stronger for more realistic situations like active locomotion, thus suppressing noise more strongly. Experiments designed to address this dichotomy could help elucidate the nature of such mechanism.

## Supporting information

**S1 Appendix.**
(DOCX)

## Author Contributions

**Conceptualization:** Björn Jörges, Laurence R. Harris.

**Data curation:** Björn Jörges.

**Formal analysis:** Björn Jörges.

**Funding acquisition:** Laurence R. Harris.

**Investigation:** Björn Jörges.

**Methodology:** Björn Jörges, Laurence R. Harris.

**Project administration:** Björn Jörges.

**Resources:** Björn Jörges.

**Software:** Björn Jörges.

**Supervision:** Laurence R. Harris.

**Validation:** Björn Jörges.

**Visualization:** Björn Jörges.

**Writing – original draft:** Björn Jörges.

**Writing – review & editing:** Björn Jörges, Laurence R. Harris.

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
