## [Decision Letter · Decision Letter 0]

25 Jan 2024

PONE-D-23-37053The Impact of Visually Simulated Self-Motion on Predicting Object MotionPLOS ONE

Dear Dr. Jörges,

Thank you for submitting your manuscript to PLOS ONE. After careful consideration, we feel that it has merit but does not fully meet PLOS ONE’s publication criteria as it currently stands. Therefore, we invite you to submit a revised version of the manuscript that addresses the points raised during the review process.

We look forward to receiving your revised manuscript.

Kind regards,

Nick Fogt

Academic Editor

PLOS ONE

Journal Requirements:

2. In your cover letter, please confirm that the research you have described in your manuscript, including participant recruitment, data collection, modification, or processing, has not started and will not start until after your paper has been accepted to the journal (assuming data need to be collected or participants recruited specifically for your study). In order to proceed with your submission, you must provide confirmation.

BJ and LRH are supported by the Canadian Space Agency (CSA) (CSA: 15ILSRA1-York). LRH is supported by a Discovery Grant from the Natural Sciences and Engineering Research Council (NSERC) of Canada (NSERC: RGPIN-2020-06093). The funders did not play any role in the study design, data collection and analysis, decision to publish, or preparation of the manuscript.

Additional Editor Comments:

Both reviewers have positive things to say about the paper. Please address the reviewer comments: specifically, please briefly address reviewer #1's comment regarding potential differences in frontal and sagittal/radial planes, and reviewer #2's comments regarding additional motion cues available when an observer is moving.

Reviewers' comments:

Reviewer's Responses to Questions

**Comments to the Author**

1. Does the manuscript adhere to the experimental procedures and analyses described in the Registered Report Protocol?

If the manuscript reports any deviations from the planned experimental procedures and analyses, those must be reasonable and adequately justified.

Reviewer #1: Yes

Reviewer #2: Yes

2. If the manuscript reports exploratory analyses or experimental procedures not outlined in the original Registered Report Protocol, are these reasonable, justified and methodologically sound?

A Registered Report may include valid exploratory analyses not previously outlined in the Registered Report Protocol, as long as they are described as such.

Reviewer #1: Yes

Reviewer #2: Yes

3. Are the conclusions supported by the data and do they address the research question presented in the Registered Report Protocol?

The manuscript must describe a technically sound piece of scientific research with data that supports the conclusions. The conclusions must be drawn appropriately based on the research question(s) outlined in the Registered Report Protocol and on the data presented.

Reviewer #1: Yes

Reviewer #2: Yes

4. Have the authors made all data underlying the findings in their manuscript fully available?

Reviewer #1: Yes

Reviewer #2: Yes

5. Is the manuscript presented in an intelligible fashion and written in standard English?

Reviewer #1: Yes

Reviewer #2: Yes

6. Review Comments to the Author

Please use the space provided to explain your answers to the questions above. (Please upload your review as an attachment if it exceeds 20,000 characters)

Reviewer #1: The research is original and the research question is of importance. Careful and comprehensive experiments and analysis were performed to address the research question. The conclusions were supported by the experiments and were presented clearly.

The authors may want to add some discussions on the design of the stimuli and explain how conclusions from studies tested on different stimuli may apply to this study. In this study, a lateral self-motion is simulated and the object motion is parallel to the observer motion. In other studies of flow parsing, a radial flow (e.g. Warren & Rushton, 2007; Dokka et al., 2015) or a rotational (e.g. Rushton & Warren, 2005) self-motion are often simulated. I am not sure if flow parsing for e.g. radial flow is the same as “flow parsing” for lateral flow because the situation seems is more complicated for a radial flow. It is possible in a simple situation like in this study, local motion contrast (Niehorster & Li, 2017) may play more role than flow parsing. Perception of object motion when the object motion is parallel to the self-motion might also be a special case because in real-life, we may never have to parse the flow if the object is in parallel as there is no danger of the object hitting us.

Reviewer #2: In the current study, the authors examined visually induced self-motion on perception of object motion, including speed and time for collision. The main finding is that the effects were largely expected based on the consistency between the self-motion direction and object-motion direction, except for a few conditions like the speed estimation on the same directions. Overall the experiments are clearly done. The results are clear, and clear explanations are provided.

1) The main concern is that there is no real self-motion cues such as vestibular, efference copy signals are provided. It is not too surprising that visual alone would cause bias due to non-perfect flow parsing. Could these experiments be applied during VR when locomotion is accompanied?

2) What about the effects when the visual flow is full of the display, instead of only on the ground?

7. PLOS authors have the option to publish the peer review history of their article (what does this mean?). If published, this will include your full peer review and any attached files.

Reviewer #1: No

Reviewer #2: No

---

## [Author Response · Author response to Decision Letter 0]

30 Jan 2024

Reviewer #1: 

The authors may want to add some discussions on the design of the stimuli and explain how conclusions from studies tested on different stimuli may apply to this study. In this study, a lateral self-motion is simulated and the object motion is parallel to the observer motion. In other studies of flow parsing, a radial flow (e.g. Warren & Rushton, 2007; Dokka et al., 2015) or a rotational (e.g. Rushton & Warren, 2005) self-motion are often simulated. I am not sure if flow parsing for e.g. radial flow is the same as “flow parsing” for lateral flow because the situation seems is more complicated for a radial flow. It is possible in a simple situation like in this study, local motion contrast (Niehorster & Li, 2017) may play more role than flow parsing. Perception of object motion when the object motion is parallel to the self-motion might also be a special case because in real-life, we may never have to parse the flow if the object is in parallel as there is no danger of the object hitting us.

RESPONSE: Lateral, rotational and forwards/backwards self-motion are indeed fairly different in terms of the geometry and in terms of their ecological usefulness. Nonetheless, findings are fairly consistent in that visually simulated self-motion alone usually leads to incomplete flow parsing and the corresponding biases in object motion judgements. We have added a section in the discussion to address these questions.

Reviewer #2: 

1) The main concern is that there is no real self-motion cues such as vestibular, efference copy signals are provided. It is not too surprising that visual alone would cause bias due to non-perfect flow parsing. Could these experiments be applied during VR when locomotion is accompanied?

RESPONSE: The finding that visual input alone led to incomplete flow parsing is, as the reviewer notes rightly, unsurprising. We have shown this before, as well as a series of other studies. The main focus of this study was to investigate the link between incomplete flow parsing for a more perception-oriented task and a more action oriented (prediction task). Given our results, it would seem to us that the most interesting hook for future studies (with vestibular and/or active observer motion) lies in our observation of lower-than-expected variability in the prediction task for visual observer motion, which motivated us to posit a potential compensatory mechanism. It would be extremely interesting how this effect holds up when there are more self-motion cues available. (Dokka et al., for example, found that increased flow parsing leads to increased noise in object motion judgements; on the other hand, if there is indeed a compensatory mechanism, it might be stronger for more realistic situations.) We have added these thoughts in the conclusions section of the paper as possible future directions for this line of research.

2) What about the effects when the visual flow is full of the display, instead of only on the ground?

RESPONSE: It has been shown (Niehorster & Li, 2017) that flow parsing can decrease when there is no local information available to the observer. This is the case in our study, so we would expect flow parsing to be somewhat more complete if local optic flow information were also available to the observer.

---

## [Editor Report · Decision Letter 1]

6 Feb 2024

The Impact of Visually Simulated Self-Motion on Predicting Object Motion

PONE-D-23-37053R1

Dear Dr. Jörges,

We’re pleased to inform you that your manuscript has been judged scientifically suitable for publication and will be formally accepted for publication once it meets all outstanding technical requirements.

Kind regards,

Nick Fogt

Academic Editor

PLOS ONE

Additional Editor Comments (optional):

Thank you for your responses to the reviewer comments.

---

## [Editor Report · Acceptance letter]

12 Feb 2024

PONE-D-23-37053R1 

PLOS ONE

Dear Dr. Jörges, 

I'm pleased to inform you that your manuscript has been deemed suitable for publication in PLOS ONE. Congratulations! Your manuscript is now being handed over to our production team.

Kind regards, 

on behalf of

Dr. Nick Fogt 

Academic Editor

PLOS ONE